# Sphingosine-1-Phosphate (S1P) Lyase Inhibition Aggravates Atherosclerosis and Induces Plaque Rupture in *ApoE^−/−^*  Mice

**DOI:** 10.3390/ijms23179606

**Published:** 2022-08-24

**Authors:** Petra Keul, Susann Peters, Karin von Wnuck Lipinski, Nathalie H. Schröder, Melissa K. Nowak, Dragos A. Duse, Amin Polzin, Sarah Weske, Markus H. Gräler, Bodo Levkau

**Affiliations:** 1Institute for Molecular Medicine III, University Hospital Düsseldorf, University of Düsseldorf, 40225 Düsseldorf, Germany; 2Division of Cardiology, Pulmonology, and Vascular Medicine, Heinrich Heine University Medical Center Düsseldorf, 40225 Düsseldorf, Germany; 3Department of Anesthesiology and Intensive Care Medicine, Center for Sepsis Control and Care and Center for Molecular Biomedicine, University Hospital Jena, 07743 Jena, Germany

**Keywords:** sphingosine-1-phosphate, atherosclerosis, plaque rupture, vascular biology, cholesterol efflux

## Abstract

Altered plasma sphingosine-1-phosphate (S1P) concentrations are associated with clinical manifestations of atherosclerosis. However, whether long-term elevation of endogenous S1P is pro- or anti-atherogenic remains unclear. Here, we addressed the impact of permanently high S1P levels on atherosclerosis in cholesterol-fed apolipoprotein E-deficient (*ApoE^−/−^*) mice over 12 weeks. This was achieved by pharmacological inhibition of the S1P-degrading enzyme S1P lyase with 4-deoxypyridoxine (DOP). DOP treatment dramatically accelerated atherosclerosis development, propagated predominantly unstable plaque phenotypes, and resulted in frequent plaque rupture with atherothrombosis. Macrophages from S1P lyase-inhibited or genetically deficient mice had a defect in cholesterol efflux to apolipoprotein A-I that was accompanied by profoundly downregulated cholesterol transporters ATP-binding cassette transporters ABCA1 and ABCG1. This was dependent on S1P signaling through S1PR3 and resulted in dramatically enhanced atherosclerosis in *ApoE^−/−^/S1PR3^−/−^* mice, where DOP treatment had no additional effect. Thus, high endogenous S1P levels promote atherosclerosis, compromise cholesterol efflux, and cause genuine plaque rupture.

## 1. Introduction

Sphingosine-1-phosphate (S1P) plays important roles in cardiovascular physiology and disease. In humans, plasma S1P concentrations have been associated with clinical manifestations of atherosclerosis such as coronary (CAD) and peripheral (PAD) artery disease and myocardial infarction [1,2,3]. Plasma S1P is contained mainly in high-density lipoproteins (HDL) and contributes to several beneficial HDL functions [1,4]. Accordingly, we have shown that HDL-contained S1P (HDL-S1P) is reduced in CAD and responsible for reduced vasodilation and anti-inflammation by CAD-HDL [5,6]. Importantly, therapeutic S1P loading of dysfunctional CAD-HDL fully restored its beneficial functions [7]. Therefore, we have defined reduced HDL-S1P as a novel marker of HDL dysfunction associated with CAD and atherosclerosis [8]. Recently, we have also identified endogenous S1P production by sphingosine kinase 2 in macrophages to be involved in cholesterol efflux to apolipoprotein A-I, the initial event in reverse cholesterol transport [9].

S1P has numerous effects on virtually all cell types involved in the pathogenesis of atherosclerosis [10]. However, no information is available about the effects of long-term raising or lowering of S1P concentrations on the development of atherosclerosis in vivo. Pharmacological studies using sphingosine kinase inhibitors have shown no changes in the extent of atherosclerosis in LDL receptor-deficient mice (*LDLR^−/−^)* despite a reduction in plasma S1P concentrations [11]. Transplantation of S1P lyase-deficient bone marrow to *LDLR^−/−^* mice resulted in a moderate decrease in plaque size in the aortic root over 4 weeks [12]. In contrast, long-term administration of pharmacological S1P analogs such as fingolimod (FTY720) has reduced atherosclerosis in apolipoprotein E deficient mice (*ApoE^−/−^*) and *LDLR^−/−^* in some [13,14] but not all studies [15]. The causal mechanisms behind such observations have remained unclear due to the manifold and sometimes opposing effects of S1P in the context of atherosclerosis on monocytes/macrophages and lymphoid, endothelial, dendritic, and vascular smooth muscle cells. S1P exerts many of its effects through S1P receptors (S1PR). Thus, several S1PR agonists and antagonists along with mice deficient for individual S1PR have been studied with respect to atherosclerosis. However, the results are complex and at times difficult to reconcile: while a pharmacological S1PR1 antagonist had no effect in *LDLR^−/−^* mice [16], an S1PR1 agonist reduced atherosclerosis [17]; endothelial-specific S1PR1 deletion enhanced [18], global S1PR2 deficiency suppressed [19], whereas S1P3R had no effect on atherosclerosis albeit altered plaque composition [20]. Of note, the presence or absence of a high-cholesterol diet in these studies apparently affected the extent of S1P effects exerted on atherosclerosis [13,14,15,16,17,20]. S1P lyase-deficient mice have altered liver lipid metabolism [21], and their extremely short (weeks) life span [22] precludes long-term metabolic or atherosclerosis studies.

The aim of the current study was to overcome this by using pharmacological S1P lyase inhibition in combination with a Western-type diet in *ApoE^−/−^* mice to investigate the effect of long-term S1P elevation on the development of atherosclerosis. We have observed dramatic effects on plaque size, composition, and stability leading to plaque rupture and atherothrombosis in this model along with alterations in macrophage cholesterol content and efflux, and we have identified the responsible S1P receptor in vivo.

## 2. Results

### 2.1. Pharmacological Inhibition of the S1P Lyase Aggravated Atherosclerosis Development in ApoE^−/−^ Mice

To assess the role of S1P lyase inhibition on atherosclerosis, *ApoE^−/−^* mice were fed three different types of diet: regular Western-type diet (Western), Western diet without vitamin B6 (Western-B6), and Western diet together with oral administration of 5 mg/kg/day 4-deoxypyridoxine (DOP), a pharmacological inhibitor of the S1P lyase, via the drinking water (Western-B6+DOP). The lack of Vitamin B6 in the diet was employed to accentuate the inhibitory effect of DOP as both compete for the same binding site on the pyridoxal-5′-phosphate binding domain of the enzyme [23,24]. Western diet without vitamin B6 did not differ from Western diet with respect to atherosclerosis development allowing us to concentrate on the two other groups. Atherosclerosis development was analyzed at 6 and 12 weeks always at two anatomical sites and by two different methods (*en face* lipid staining of the aorta and total plaque volume calculation in the entire brachiocephalic artery (BCA) [14]).

Supplementation of DOP with the diet resulted in a remarkable 3-fold increase in lipid deposition in the aorta after 6 weeks and a 2-fold increase, respectively, after 12 weeks, as compared to Western diet (Figure 1A,B). Plaque volume in the BCA was increased 4-fold after 6 weeks and 3-fold after 12 weeks, respectively, in the DOP-treated group compared to Western diet (Figure 2A). The efficiency of S1P lyase inhibition was confirmed by the 3-fold higher plasma S1P concentration in the DOP-treated group as measured by mass spectrometry after 6 and 12 weeks (Figure 1C). S1P levels in single aortae were increased even further (10-fold) by DOP (Figure 1D).

In addition to S1P, there was also an elevation of the plasma levels of ceramide-16 (2-fold) and phosphatidylcholine (1.4-fold) with DOP, whereas sphingosine and sphingomyelin concentrations were not altered (Table 1).

### 2.2. Characteristics of Atherosclerotic Lesions of S1P Lyase-Inhibited ApoE^−/−^ Mice

Atherosclerotic lesions in the BCA of mice on a Western-B6+DOP diet had substantially larger (~58%) necrotic cores than mice on a Western diet (Figure 2B). Plaque smooth muscle cell content was reduced by 52% and that of collagen by 72% (Figure 2B), respectively. Plaque macrophage content was similar between the groups (Figure 2B). Reduced collagen and smooth muscle cell content is generally perceived as an indication of reduced plaque stability in humans and mice, although real plaque rupture and thrombosis do not occur in *ApoE^−/−^* mice on a Western-type diet at this age.

### 2.3. Lipoprotein and Leukocyte Changes after S1P Lyase Inhibition in Cholesterol-Fed ApoE^-/-^ Mice

As atherosclerosis in mice is intimately linked to plasma cholesterol levels, we measured those and observed a ~2-fold elevation in plasma cholesterol in Western-B6+DOP-treated *ApoE^−/−^* mice compared to the Western group both after 6 and 12 weeks of treatment (Figure 3A). This increase was due to the elevation of the cholesterol contained in the LDL/VLDL fraction, whereas HDL-cholesterol was not altered (Figure 3B). S1P lyase inhibition is known to induce characteristic changes in lymphocyte subpopulations due to its disruption of S1P gradients between plasma and tissues [25]. Accordingly, DOP-treated *ApoE^−/−^* mice exhibited a pronounced decline in both CD4^+^ and CD8^+^ T-lymphocytes in the peripheral blood over the entire treatment period (Figure 3C,D). Interestingly, monocyte numbers (F4/80^+^CD115^+^) were higher in the Western + DOP group compared to the Western group, with inflammatory (Gr-1^high^) and patrolling (Gr-1^low^) monocytes being increased to the same extent (Figure 3E,F).

### 2.4. Plaque Rupture in Atherosclerotic Lesions of S1P Lyase-Inhibited ApoE^−/−^ Mice

Plaque rupture is virtually never observed in *ApoE^−/−^* mice on a regular Western-type diet composition of this duration and at this age. Therefore, it was surprising to frequently observe genuine plaque rupture with appositional atherothrombosis in Western-B6+DOP-treated *ApoE^−/−^* mice: 5 out of 18 brachiocephalic arteries (28%) showed intraluminal thrombosis protruding into the arterial lumen and emerging from a clearly identifiable fibrous cap rupture (defined as visible cap discontinuity; Figure 4 and Appendix A). All thrombi were of the mixed type consisting of fibrin, platelets, and red blood cells indicating recent formation (Figure 4A–E). In contrast, no thrombi at all were present in the Western group (Figure 4F). Two morphological criteria consistent with and described in murine plaque rupture [26] were also present in the plaques of DOP-treated mice: 9 out of 18 plaques (50%) showed fibrin deposition within the lesion and 5 (28%) featured erythrocytes inside complex plaque architecture (‘intraplaque hemorrhage’; Figure 4F). In contrast, no fibrin deposition was present in any of the plaques of the Western group, and only a single lesion had a small intralesional erythrocyte spot (Figure 4F and Appendix A).

### 2.5. Profound Changes in Macrophage Sphingolipid and Cholesterol Content but Unaltered Macrophage Polarization, Inflammation and Efferocytosis

Western-B6+DOP diet resulted in profoundly altered sphingolipid content of macrophages already after 1 week and persisting after 6 weeks (Table 1). Elicited peritoneal macrophages from mice treated with Western-type-B6 + DOP for 1 week featured 1000-fold increased S1P, 100-fold sphingosine, 15-fold ceramide-16, and 2-fold sphingomyelin and phosphatidylcholine, respectively (Table 1). Six weeks after treatment, S1P, sphingosine, and ceramide-16 concentrations in macrophages were still increased 800-fold, 15-fold, and 3-fold, respectively, while those of sphingomyelin and phosphatidylcholine were similar to the Western group (Table 1).

We have also observed a profoundly increased free cholesterol content in macrophages from Western-B6+DOP-treated mice (Figure 5A). As free cholesterol is an important player in several macrophage functions, we analyzed macrophage polarization, inflammatory potential, and efferocytosis. However, none of these parameters were different: gene expression of M1/M2 macrophage markers such as iNOS and arginase and that of numerous cytokines, chemokines, and inflammatory genes was similar (Appendix A); stimulation with LPS/IFN resulted in an equipotent induction of TNFα and iNOS gene expression (Figure 5B,C), and the phagocytosis of apoptotic lymphocytes was unaltered (Figure 5D).

### 2.6. Genetic Deficiency and Pharmacological Inhibition of the S1P Lyase Downregulate Macrophage ABCA1 and ABCG1 Gene Expression and Impair Cholesterol Efflux to ApoA-I

Gene expression analysis revealed that two major cholesterol transporters involved in reverse cholesterol transport—the ATP-binding cassette transporters A1 (ABCA1) and G1 (ABCG1)—were downregulated by more than 90% in macrophages from Western-B6+DOP-treated *ApoE^−/−^* mice compared to the Western group after 6 weeks (Figure 6A,B). Genetic S1P lyase deletion affected the expression of macrophage cholesterol transporters in a similar manner: macrophages from inducible S1P lyase knockout mice, where the Cre recombinase is driven by an inducible ß-actin promoter (actb-CreERT2) [27,28], displayed a 60–70% reduction in both ABCA1 and ABCG1 gene expression compared to littermate controls (Figure 6E). To assess the functional relevance of the downregulated cholesterol transporters, we measured [^3^H]-cholesterol efflux to human apolipoprotein A-I (ApoA-I) in macrophages from control and inducible KO S1P lyase mice. We performed these experiments both under basal conditions and after additional treatment with DOP in vitro. Inducible S1P lyase-KO macrophages exhibited a 47% reduction in cholesterol efflux to ApoA-I under basal conditions compared to littermate controls (Figure 6F). DOP treatment for 24 h inhibited [^3^H]-cholesterol efflux to ApoA-I in control macrophages by 50–60% but failed to reduce the already compromised efflux in inducible S1P lyase-KO macrophages any further (Figure 6F). In agreement with the in vivo data, DOP treatment in this setting suppressed ABCA1 and ABCG1 gene expression by 70–80% in control macrophages (Figure 6C,D).

### 2.7. Attenuated Cholesterol Efflux in S1PR3 Deficient Macrophages and Enhanced Atherosclerosis in Cholesterol-Fed ApoE^−/−^/S1PR3^−/−^ Mice

We have previously reported that the S1P receptor S1PR3 is crucially involved in macrophage cholesterol efflux to ApoA-I in response to stimulation with the LXR agonist 22(R)-hydroxycholesterol [9]. Here, we tested if stimulation with S1P affected cholesterol efflux and whether S1PR3 played a role. Indeed, S1P suppressed cholesterol efflux to ApoA-I by 25% in *S1PR3^+/+^* macrophages (Figure 7A). In comparison, *S1PR3^−/−^* macrophages had already a 28% reduced cholesterol efflux to ApoA-I and, most importantly, were resistant to any further decrease by S1P (Figure 7A). Furthermore, ABCA1 gene expression was reduced in *ApoE^−/−^/S1PR3^−/−^* macrophages, and was downregulated by S1P in *ApoE^−/−^/S1PR3^+/+^* but not *ApoE^−/−^/S1PR3^−/−^* macrophages (Figure 7E).

To explore the significance of these findings for atherosclerosis, we performed a new series of experiments to compare atherosclerotic lesion development in *ApoE^−/−^/S1PR3^−/−^* mice versus *ApoE^−/−^/S1PR3^+/+^* mice on a Western diet for 12 weeks with and without DOP. Remarkably, plaque volume was dramatically increased (22.8-fold) in *ApoE^−/−^/S1PR3^−/−^* compared to *ApoE^−/−^/S1PR3^+/+^* mice without DOP (Figure 7B,C). DOP treatment increased plaque volume in *ApoE^−/−^/S1PR3^+/+^* mice as expected from our initial study but did not increase the already substantial plaque burden in *ApoE^−/−^/S1PR3^−/−^* mice (Figure 7B,C). Plasma cholesterol in untreated *ApoE^−/−^/S1PR3^−/−^* was already as high as in *ApoE^−/−^/S1PR3^+/+^* on DOP and increased even further with DOP (Figure 7D).

## 3. Discussion

In several clinical studies, alterations in plasma S1P have been associated with different clinical manifestations of atherosclerosis such as CAD and myocardial infarction [1,2,3]. However, no causal relationships or underlying reasons have been defined. Some of the described plasma S1P alterations can be of chronic nature such as those in stable CAD or occur acutely as in myocardial infarction or PCI-related myocardial ischemia [5,29]. S1PR are present and functional on virtually all cells relevant to the pathogenesis of atherosclerosis, and numerous potentially atherosclerosis-related S1P functions have been described. However, the effects of chronic S1P elevation on atherosclerosis in vivo have remained unexplored. To our knowledge, our study is the first to examine atherosclerosis progression in the presence of high endogenous S1P levels in the animal model of cholesterol-fed *ApoE^−/−^* mice over a period of up to 12 weeks. The only other study on this topic has observed moderately reduced atherosclerosis in hematopoietic *S1P lyase*/*LDLR^−/−^* chimeric mice after 4 weeks [12]. The mechanism has remained obscure particularly as serum cholesterol levels were reduced, female *LDLR^−/−^* mice are known to develop minimal atherosclerosis, and 4 weeks are far too short for mice to develop substantial disease, especially with the preceding irradiation further slowing it. In contrast, our study has demonstrated accelerated and aggravated atherosclerotic plaque development with increased levels of plasma cholesterol. It is well established that hypercholesterolemia alone is sufficient to induce atherosclerosis in mouse models and serves as an established cause of monocytosis [30,31]. Indeed, both hypercholesterolemia and monocytosis are present in our model and could have contributed to the observed phenotype. A further distinct and unique plaque feature we have observed is genuine plaque rupture with appositional intraluminal atherothrombosis. This is remarkable as plaque rupture usually does not occur in *ApoE^−/−^* mice on regular Western-type diets and has been seen occasionally as the result of diet-induced extreme hypercholesterolemia over long time periods [32,33]. Even there, appositional atherothrombosis was extremely rare. In contrast, it occurred in almost 30% of all DOP-treated mice in our study. Additional confirmation of plaque instability was provided by the presence of key characteristics associated with plaque rupture in the literature such as intra-plaque fibrin deposition (present in 50% of all DOP-treated mice in our study) and intraplaque hemorrhage. A suitable explanation for this phenotype may be the combination of large plaque necrotic cores, extremely thin fibrous caps, and the substantially reduced collagen and smooth muscle cell content that all together result in plaque instability. Our observation of frequent genuine plaque rupture with atherothrombosis in S1P lyase-inhibited mice is unique among the mouse models of atherosclerotic disease and very reminiscent of the human situation as the cause of myocardial infarction. Thus, the phenotype we have observed should be considered in human studies on plaque rupture and myocardial infarction in the context of elevated plasma S1P levels.

Mechanistically, an important hint to understanding the disturbance of cholesterol homeostasis occurring in the presence of elevated S1P concentrations in our study came from the observation that macrophages from DOP-treated mice contained high amounts of free cholesterol. The second hint was the dramatic downregulation of two key cholesterol transporters, ABCA1 and ABCG1, the functional relevance of which was asserted by reduced cholesterol efflux. ABCA1 and ABCG1 are well known to play a central role in cholesterol efflux from macrophages [34]. ABCA1 has been shown to be causally involved in reverse cholesterol transport [35], and combined deletion of ABCA1 and ABCG1 in mouse macrophages dramatically impaired cholesterol efflux to ApoA-I, accelerated atherosclerosis, and promoted foam cell accumulation [36,37,38,39]. Our data on downregulated ABC transporters in mice with chronic pharmacological lyase inhibition in vivo were confirmed by in vitro studies with acute lyase inhibition as well as genetic lyase deficiency. In each case, ABCA1 gene expression and cholesterol efflux were reduced. Nevertheless, one particular characteristic should be stressed in relation to other studies on ABCA1 deficiency in vivo in the context of atherosclerosis. Although we, too, have observed intracellular cholesterol accumulation and impaired ABCA1-mediated cholesterol efflux in S1P lyase-inhibited or -deficient macrophages, the accompanying hypercholesterolemia and acceleration of atherosclerosis are uncommon to global ABCA1 deficiency either on the *ApoE^−/−^* or *LDLR^−/−^* background [40]. In fact, plasma cholesterol is lower there due to the reduction in apolipoprotein B-containing lipoproteins as well as the absence of HDL and atherosclerosis unaffected [40]. Somewhat closer to our model is the phenotype of macrophage-specific ABCA1 deficiency on the *ApoE^−/−^* or *LDLR^−/−^* backgrounds, where atherosclerosis is enhanced and plasma cholesterol is normal [40,41]. In our case, the increase in plasma cholesterol was due to higher LDL/VLDL cholesterol but normal HDL cholesterol suggesting that additional mechanisms are driving plasma cholesterol elevation. Indeed, global S1P lyase knockout mice [22] also have higher plasma cholesterol and hepatic disturbances of lipid homeostasis [42]. A hint that S1PR may be involved in the process relates to the observation that FTY720 treatment increased plasma cholesterol in *ApoE^−/−^* mice on a regular chow diet [15].

The second major finding of this study is the identification of S1P and S1PR3 receptors as regulators of ABCA1-mediated cholesterol efflux. This is extremely intriguing as ABCA1 itself has been identified as an S1P exporter in astrocytes in a process that involved ABCA1-mediated HDL-like lipoprotein formation [43]. As ABCA1-mediated cholesterol efflux is the quintessential function of ABCA1, a scenario becomes imaginable where S1P is exported from the cell by a bona fide cholesterol exporter and, by engaging cell surface S1P receptors, inhibits the same exporter in a feedback manner. Indeed, we have previously demonstrated that macrophage ABCA1 expression and cholesterol efflux are regulated by endogenous sphingosine kinase 2-derived S1P [9]. According to this scenario, ABCA1-mediated cholesterol efflux from macrophages generates lipidated ApoA-I particles that incorporate simultaneously secreted S1P; something very similar has been already shown for astrocytes [43]. S1P contained in such particles would then limit further cholesterol efflux through S1PR3 in a feedback regulatory manner. The fact that S1PR3-deficient macrophages had lower ABCA1 gene expression, reduced cholesterol efflux, and did not respond to S1P stimulation by any additional decrease, suggests that continuous, tonic S1PR3 signaling is required for the maintenance of basal cholesterol efflux. ABCA1 gene expression is under the transcriptional control of liver X receptors (LXRs) [44], and we have observed LXR gene expression to be suppressed (5-fold) in macrophages after 6 and 12 weeks of DOP treatment (Appendix A), mirroring the regulation of ABCA1. Nuclear signaling by G-protein-coupled receptors in contrast to cell membrane signaling is well documented in the literature, and there is a precedence of cross-talk with classical nuclear receptors as, e.g., modified forms of lysophosphatidic acid and S1P bind to and activate peroxisome proliferator-activated receptor gamma [45,46]. We also compared cholesterol efflux in Western and Western-B6+DOP macrophages after 6 weeks of diet and observed similar absolute but lower relative efflux taking into account the two-fold higher cholesterol content of the latter (data not shown). The in vivo consequences of S1PR3 deficiency for atherosclerosis are in line with its positive role in cholesterol efflux. The lack of plaque rupture despite the further exaggerated phenotype in *ApoE^−/−^/S1PR3^−/−^* may be due to the plaque-stabilizing effect of S1PR3 that we previously described [20].

There certainly are other effects of high S1P levels and deregulated S1P gradients as a result of S1P lyase inhibition that may also be contributing to the atherosclerosis phenotype observed here. Dramatic changes in lymphocyte subpopulations, alterations in leukocyte rolling, and impact on macrophage recruitment by S1P are only a few to be named. Furthermore, lack of S1P lyase also impacts the levels of several other sphingolipid metabolites that may also play a role. Nevertheless, the probably most important contribution of our study is the identification of a direct functional connection between cholesterol efflux and S1P receptor signaling in macrophages with an impact on atherosclerosis and its complications. This generates an exciting new area of research that is relevant to both cholesterol and sphingolipid metabolism and associated diseases.

## 4. Materials and Methods

### 4.1. Mice

C57Bl6J and *ApoE^−/−^* mice were obtained from the Jackson Laboratory. Inducible S1P lyase knockout mice, in which the Cre recombinase is driven by an inducible ß-actin promoter (actb-CreERT2) yielding *Sgpl1^Flox/Flox^;Cre^+/−^* (=inducible KO) and *Sgpl1^Flox/Flox^;Cre^−/−^* (=control) littermates [27,28], were from Novartis, courtesy of Dr. Andreas Billich. *ApoE^−/−^* mice were fed a high cholesterol diet (21% fat, 0.15% cholesterol, 19.5% casein; Western type) and Western type without vitamin B6 supplementation for 6 and 12 weeks. 4-deoxypyridoxine (DOP) was administered with the drinking water at a dose of 5 mg/kg/day as described [25].

### 4.2. Hematology and Flow Cytometry

Blood was collected from the retroorbital plexus into EDTA tubes at the time of sacrifice. Total and differential blood counts were measured using an automated Animal ABC Coulter Counter (VetABC, scil animal care company, Viernheim, Germany). For analysis of leukocytes in blood and peritoneal lavage by flow cytometry, red blood cells were lysed using ACK buffer (0.15 M NH_4_Cl; 1 mM KHCO_3_; 0.1 mM Na_2_EDTA; pH 7.2–7.4) for 2 min at room temperature. The cells were stained with fluorochrome-conjugated combinations of the following monoclonal antibodies: CD4 (Acris, Herford, Germany), CD8, CD45, B220, Gr-1 (all from BD Biosciences, Heidelberg, Germany), F4/80 (AbD Serotec, Düsseldorf, Germany), CD3e, CD11b (Miltenyi Biotec, Bergisch Gladbach, Germany), CD115, CD25, CD44 and CD62L (eBioscience, Frankfurt, Germany) and were analyzed using a Gallios flow cytometer (Beckman Coulter, Krefeld, Germany). The size of each cell population was determined as a product of the total white blood cell count as well as total organ or peritoneal cell numbers and the percentage of the respective cell population.

### 4.3. Plasma Lipid and Lipoprotein Analysis

Lipid analysis was performed using an assay from Analyticon Biotechnologies AG, Lichtenfels, Germany. HDL and LDL/VLDL quantification was performed using a colorimetric quantification assay (BioVision, Milpitas, CA, USA).

### 4.4. Lesion Volume

Quantification of atherosclerotic lesion volume in the brachiocephalic artery (BCA) was performed as previously described [14]. After sacrifice, mice were perfusion-fixed with 4.5% neutral-buffered formaldehyde (PFA) and the BCA was dissected from the bifurcation off the aortic arch to the branching point of the right subclavian and common carotid artery, and embedded in paraffin. The entire BCA was serially sectioned in 5 µm sections. Beginning from a random start site within the first 75 µm, every 15th section was stained with hematoxylin and eosin. Images were captured with a Zeiss Axio Cam, and lesion area was quantified using AxioVision 4.6 software (Carl Zeiss Vision GmbH, Jena, Germany). The volume of the BCA lesion was determined using the Cavalieri stereological method [Σ (lesion area) × (distance; 75 µm)]. All analyses were performed without knowledge of the tissue source.

### 4.5. Oil Red O Staining of Aortae

Fixed aortae were carefully cleaned from all adventitial fat, washed in isopropanol/H_2_O (2:1) for 5 min, and stained with 0.3% Oil Red O for 90 min in 24 well plates at RT. After two washes with isopropanol/H_2_O (2:1) for 2 min, the aortae were transferred to 24-well plates and stored in PBS at 4 °C. For *en face* quantification of lipids, the aortae were placed on a black silicone plate and cut open starting at the lesser curvature. The aortae were then pinned down with acupuncture needles, and pictures were taken with a Nikon DS-2Mv camera system. The lipid-containing plaque area was determined as the percent of Oil Red O-stained area from the total aortic surface area using the AxioVision 4.6 software.

### 4.6. Immunohistochemistry

Immunostaining was performed on sections adjacent to that used for quantification of lesion area and using the following antibodies: rat anti-mouse Mac-2 (Cedarlane Laboratories, Hornby, Canada) and mouse anti-human α smooth muscle-actin (αSMA, Dako, Glostrup, Denmark). Primary antibodies were incubated for 1 h at room temperature in 3% serum matched to the species of the secondary antibodies followed by biotinylated secondary antibodies for 30 min and horseradish peroxidase-conjugated streptavidin for 45 min. Visualization was performed with diaminobenzidine and counterstaining of nuclei with hematoxylin. Necrotic cores were determined by using H&E staining. Collagen content was assessed using picrosirius red staining and polarized light. Immunostaining for every marker was performed in a single procedure for all sections. An isotype-matched antibody was used on adjacent sections as a negative control. Automated color segmentation of images was performed and a color threshold of chromogen stain was defined using the negative control, above which the stain was considered positive (Zeiss AxioVision Rel. 4.6 software). The area containing all positive pixels (µm^2^) was expressed as a percentage of the total plaque area. We used a modified version of Carstairs staining to distinguish between platelets and fibrin as described [26,47]. Tissue sections were first hydrated and stained in 5% ferric ammonium sulphate for 5 min, rinsed in running tap water, and stained in Mayer’s hematoxylin for 5 min. After washing in tap water, tissue sections were stained in picric acid-orange G solution for 40 min and rinsed once in distilled water. The sections were then stained in Ponceau fuchsin solution for 5 min, rinsed in distilled water, and differentiated with 1 % phosphotungstic acid until the muscle was red and the background pale pink, respectively, and rinsed again in distilled water. Finally, sections were stained with 1% aniline blue solution for 15 min and rinsed in several changes of distilled water followed by dehydration, clearing in xylene, and coverslipping. The fixation time is important for the exact color of fibrin and platelets [47]: in sections from tissue fixed for less than 48 h, fibrin is orange to orange-red and platelets are light gray. In tissue fixed for more than 48 h, fibrin is bright red and platelets gray-blue to navy-blue. Collagen appears bright blue at all times, muscle is red and red blood cells are clear yellow after 48 h of fixation; they are variably stained (yellow, red, or green) in tissue that was fixed for less than 48 h.

### 4.7. Isolation of Peritoneal Macrophages and In Vitro Gene Expression Studies

Peritoneal macrophages were collected after intraperitoneal injection of aged, sterile 3% thioglycollate (Sigma-Aldrich, Munich, Germany) for 96 h. To isolate them and obtain peritoneal fluid, 4 mL of pre-warmed PBS was injected into the abdominal cavity and carefully aspirated. After centrifugation for 5 min at 1200 rpm and 4 °C, red blood cells were lysed using ACK buffer (0.15 M NH_4_Cl; 1 mM KHCO_3_; 0.1 mM Na_2_EDTA; pH 7.2–7.4) for 2 min at room temperature, washed twice with ice-cold PBS, counted, and directly analyzed by flow cytometry, used for gene expression studies or cholesterol efflux experiments. For gene expression studies, cells were placed in RPMI 1% FBS. After 2 h, adherent cells were washed with PBS. The medium was changed to RPMI 0.2% BSA, stimulated or not, and incubated at 37 °C for 6 h. The following stimuli were used: 4-deoxypyridoxine (DOP; 0.25 mM), sphingosine-1-phosphate (S1P; 1 µM), the sphingosine kinase inhibitor 4-[[4-(4-chlorophenyl)-2-thiazolyl]amino]-phenol (SKi), forskolin (10 µM), and isobutyl methylxanthine (IBMX; 100 µM).

### 4.8. RNA Isolation and Quantitative PCR Analysis of Gene Expression

Total RNA was isolated using the InviTrap^®^ Spin Universal RNA Mini Kit from Invitek according to the manufacturer’s protocol. Copy DNA was generated from equal amounts of RNA with the Revert Aid First Strand cDNA Synthesis Kit (Thermo Scientific, Germany). Quantitative real-time PCR was performed in a C1000^TM^ Thermal Cycler using iQ SYBR Green Supermix (BioRad, Munich, Germany) and a standard RT-PCR protocol (3 min at 95 °C, followed by 40 cycles of 20 s at 95 °C, 20 s at 55 °C, and 20 s at 72 °C, followed by 10 min at 95°C and the melting curve of incremental increases of 0.5 °C every 5 s from 65 °C to 95 °C). Relative quantification of gene expression was calculated by the 2^−^^ΔΔCT^ method using GAPDH as an endogenous reference for normalization. Primers were used from QuantiTect Primer for iNOS and TNFa (Qiagen, Hilden, Germany) and sequences from BioTez (BioTez, Berlin, Germany) as follows: ABCA1: forward 5′- GCTTGTTGGCCTCAGTTAAGG-3′; reverse 5′-GTAGCTCAGGCGTACAGAGAT-3′; ABCG1: forward 5′- CTTTCCTACTCTGTACCCGAGG-3′; reverse 5′-CGGGGCATTCCATTGATAAGG-3′.

### 4.9. Cholesterol Efflux

Peritoneal macrophages were isolated after intraperitoneal injection of aged, sterile 3% thioglycollate medium (Sigma-Aldrich, Munich, Germany). Macrophages were plated in RPMI 1640 medium (Invitrogen, Karlsruhe, Germany) supplemented with 1% fetal bovine serum (FBS) and penicillin (100 U/mL)/streptomycin (100 µg/mL) (Invitrogen), allowed to adhere for 2 h and non-adherent cells were removed by washing twice with pre-warmed sterile phosphate-buffered saline (PBS, Applichem, Darmstadt, Germany). Macrophages were loaded with 1 μCi/mL ^3^H-cholesterol (Perkin Elmer, Waltham, MA) in RPMI containing 1% FBS and an inhibitor of the acyl-CoA cholesterol acyltransferase (ACAT inhibitor; 2 µg/mL; Sandoz 58-035, Santa Cruz, CA, USA) for 24 h. Macrophages were then washed twice with pre-warmed PBS and efflux medium (αMEM with 25 mM Hepes, Invitrogen) supplemented with ACAT inhibitor and recombinant apoA-I (10 µg/mL) was added for 5h. Stimulation with 1 µM S1P was performed by adding S1P 30 min before ApoA-I. In the case of DOP, the agent was present in the medium (DOP: 0.25 mM) during loading with ^3^H-cholesterol as well as during the efflux procedure. Stimulation with forskolin/IBMX +/− S1P was performed in RPMI with 0.2% BSA and 2 µg/mL ACAT inhibitor supplemented with IBMX (100 µM) and forskolin (10 µM) in the presence or absence of 1 µM S1P for 18 h. The medium was then removed, cells washed twice with PBS, and efflux medium containing ACAT inhibitor was added. S1P (1 µM) was again added to the medium, and the cholesterol efflux assay was started 30 min later by adding ApoA-I for 5 h. The medium was then taken off and transferred to 1.5 mL tubes. Following tabletop centrifugation (5 min, 2000 rpm at room temperature; RT) to pellet cellular debris radioactivity in 100 µL of supernatant was determined by liquid scintillation counting (Beckman LS 6500). Cells were washed with PBS and intracellular lipid extraction was performed by adding hexan/isopropanol (3:2, v:v) for 30 min at room temperature. The hexan/isopropanol fraction was transferred into a 1.5 mL tube and hexan/isopropanol was added again to the cells, incubated for another 15 min at room temperature, and transferred to the same tube. After drying overnight under the hood, extracted lipids were dissolved in isopropanol/NP-40 (9:1, v:v) and subjected to scintillation analysis. Efflux per well was expressed as the percentage of counts released into the medium relative to the total amount of radioactivity initially present (counts recovered within the medium added to the counts recovered from the cells). Values obtained from control cells without ApoA-I were subtracted from all respective experimental values to correct for baseline efflux.

### 4.10. Fluorimetric Cholesterol Assay

Quantification of cholesterol using the fluorimetric cholesterol assay was performed as previously described [48]. For cholesterol loading, mouse macrophages were seeded into 6 well plates. Total cholesterol was extracted from the cells using 1 mL hexane/isopropanol (3:2, *v*:*v*). The extracts were transferred to 1.5 mL reaction tubes and dried overnight under the extractor hood. After solvent extraction cell protein was isolated from the same wells by adding 1 mL 0.2 N NaOH for 3 h at 37 °C. The plate was subsequently shaken for 5 min at room temperature. Lysates were transferred into microfuge tubes, and protein concentration was determined with a BCA kit (Pierce, Belgium). Quantification of free cholesterol was performed following exactly established protocols [48]. Fluorescence was measured at an excitation wavelength of 528 nm (filter: 528 ± 10 nm) and an emission wavelength of 590 nm (filter: 590 ± 20 nm). Total cholesterol was quantified by supplementing cholesterol esterase to the reaction. Cholesterol esters were determined by subtracting the free cholesterol values from the total cholesterol values [48]. Data for total and free cholesterol and cholesterol esters are indicated as µg per mg of cell protein. 

### 4.11. S1P Measurements

Concentrations of S1P, sphingosine, ceramide-16, sphingomyelin, and phosphatidylcholine in plasma and cells were determined by liquid chromatography coupled to tandem mass spectrometry (LC/MC/MC) according to established protocols [49]. Briefly, lipids were extracted from plasma or cell suspension by successive addition of 1 mL of methanol, 200 µL of 6 M HCl, and twice 2 mL of chloroform. Chloroform phases were retrieved by centrifugation, and chloroform was removed by vacuum-drying in a speed-vac. Subsequently, samples were dissolved in 200 µL of dioxane, 200 µL of 70mM K_2_HPO_4_, and 200 µL of 9-fluorenylmethyl chloroformate (FMOC-Cl) solution for derivatization of sphingolipids. For chromatographic detection of sphingolipids (Merck-Hitachi Elite LaChrom System, VWR, Radnor, PA, USA), 10 µL of sample was injected by cut-injection method with an injection-pump delivery rate of 1.3 mL/min into an eluent containing methanol, 70 mM K_2_HPO_4_, and H_2_O. Columns used for the separation of sphingolipids by reversed-phase HPLC were a 250 × 4.6 mm Kromasil 100-5 C18 column and a 17 × 4 mm Kromasil 100-5 C18 pre-column (CS Chromatographie Service, Langerwehe, Germany) set to 35 °C. Detection was performed with a fluorescence detector (excitation 263 nm, emission 316 nm).

### 4.12. Macrophage Phagocytosis Assay

Peritoneal macrophages were isolated 96 h after thioglycollate injection, stained with the red fluorescent dye PKH26 (Sigma-Aldrich, Darmstadt, Germany), and plated into chamber slides for 2 h. Apoptosis of Jurkat cells was induced separately by UV irradiation for 10 min. Cells were subsequently stained with the green-fluorescent dye PKH67 (Sigma-Aldrich, Darmstadt, Germany) and added to the macrophages. Phagocytosis was analyzed after 2 h by manual counting of macrophages positive for phagocytosed Jurkat cells (per 100 macrophages) using a fluorescence microscope.

### 4.13. Statistics

Statistical significance was evaluated by a two-way ANOVA followed by Bonferroni post hoc analysis, by one-way ANOVA followed by Tukey’s post hoc test, or by a two-tailed paired or unpaired Student’s *t*-test where appropriate as indicated in each figure legend. Two-sided probability values less than 0.05 were considered significant.

### 4.14. Study Approval

All experiments were approved by the Landesamt für Natur, Umwelt, und Verbraucherschutz Nordrhein-Westfalen, Germany.

## Figures and Tables

**Figure 1 ijms-23-09606-f001:**
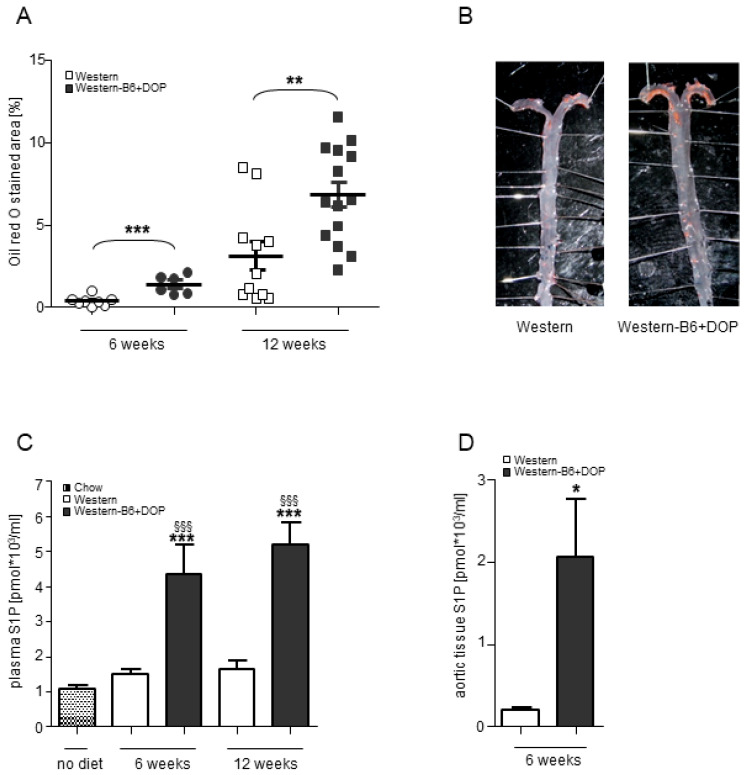
**Aggravated atherosclerotic lesion development in cholesterol-fed *ApoE^−/−^* mice after pharmacological inhibition of the S1P lyase.** *ApoE^−/−^* mice were fed Western diet or Western diet without vitamin B6 with or without DOP. (**A**) Quantification of lipid accumulation in the aortae after 6 and 12 weeks of DOP treatment as measured by *en face* staining with Oil Red O. Results are shown as mean ± SEM. Statistically significant differences were tested for using a one-way ANOVA followed by Tukey’s post hoc analysis; ** *p* < 0.01; *** *p* < 0.001. (**B**) Representative *en face* stainings of aortae after 12 weeks of DOP treatment. (**C**) Plasma S1P levels in all groups as determined by liquid chromatography coupled to tandem mass spectrometry (n = 9–20 animals per group). Results are shown as mean ± SEM. Statistically significant differences from controls were analyzed by one-way ANOVA followed by Tukey’s post hoc analysis; *** *p* < 0.001 (DOP versus ApoE-Western and ApoE-Western-B6) and ^§§§^ *p* < 0.001 (versus *ApoE^−/−^* on chow diet). (**D**) S1P concentrations in aortic tissue of mice with no Western versus Western-B6+DOP diet for 6 weeks (n = 6 and n = 7). Results are shown as mean ± SEM. Statistically significant differences from controls were analyzed by Student’s *t*-test; * *p* < 0.05.

**Figure 2 ijms-23-09606-f002:**
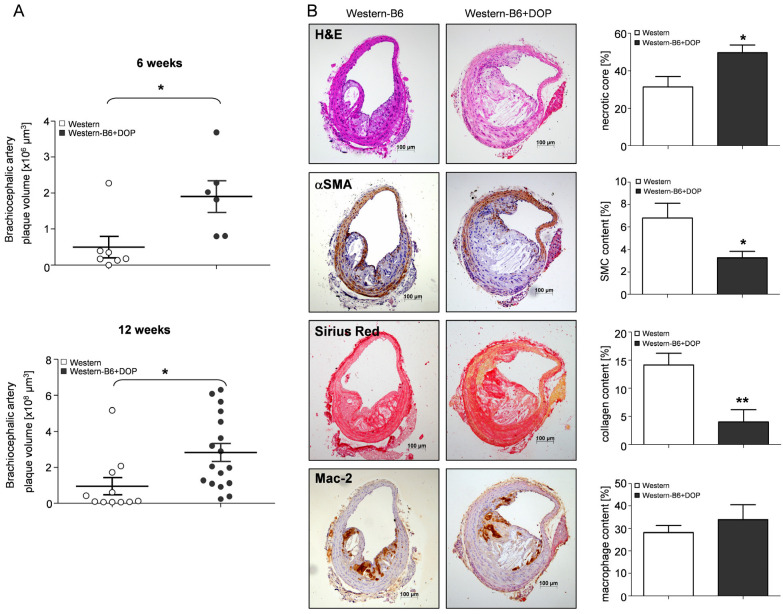
**Large necrotic cores and altered plaque composition in atherosclerotic lesions of DOP-treated *ApoE^−/−^* mice.** (**A**) Quantification of plaque volume in the entire brachiocephalic artery (BCA) after 6 and 12 weeks of DOP treatment performed using Cavalieri’s stereologic method. Results are shown as mean ± SEM. Statistical significance was assessed by Student’s *t*-test. * *p* < 0.05. (**B**) Analysis of plaque composition in the BCA by H&E staining (necrotic core) and immunostaining for smooth muscle cells (αSMA), collagen (picrosirius red) and macrophages (Mac-2). Representative examples of each staining are shown to the left and quantitative data on the right (Student’s *t*-test; * *p* < 0.05; ** *p* < 0.01). Magnification 100×.

**Figure 3 ijms-23-09606-f003:**
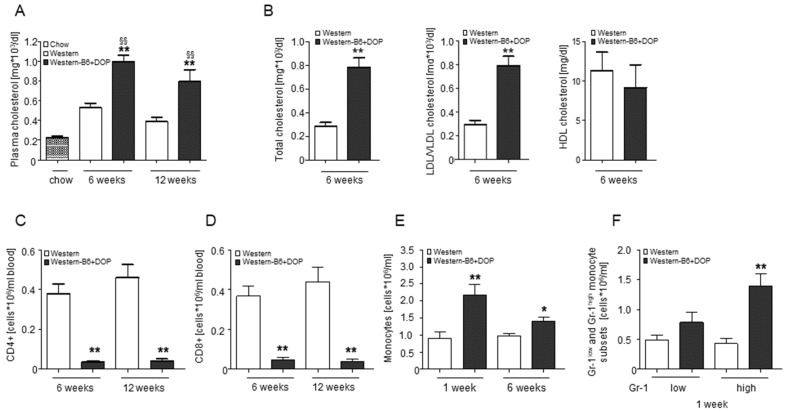
**Lipoprotein and leukocyte alterations after S1P lyase inhibition in cholesterol-fed *ApoE^−/−^* mice.** (**A**) Total plasma cholesterol levels over a period of 6 and 12 weeks. Statistically significant differences were analyzed by one-way ANOVA followed by Tukey’s post hoc test. ** *p* < 0.01 (DOP compared to Western) and ^§§^ *p* < 0.01 (compared to normal chow-fed *ApoE^−/−^*). (**B**) Quantification of total cholesterol, LDL/VLDL cholesterol, and HDL cholesterol in plasma samples of mice fed a Western versus Western-B6+DOP diet after 6 weeks. Statistically significant differences (Student’s *t*-test) are indicated as ** *p* < 0.01. All results are shown as mean ± SEM. (**C**,**D**) Severe lymphopenia as observed by the decline in CD4+ T-lymphocytes and CD8+ T-lymphocytes after DOP treatment. (**E**) Monocytosis and (**F**) quantification of F4/80^+^CD115^+^Gr-1 high and low monocyte subsets in peripheral blood. Results are shown as mean ± SEM. Statistically significant differences from controls were analyzed by Student’s *t*-test; * *p* < 0.05; ** *p* < 0.01 Western compared to Western-B6+DOP).

**Figure 4 ijms-23-09606-f004:**
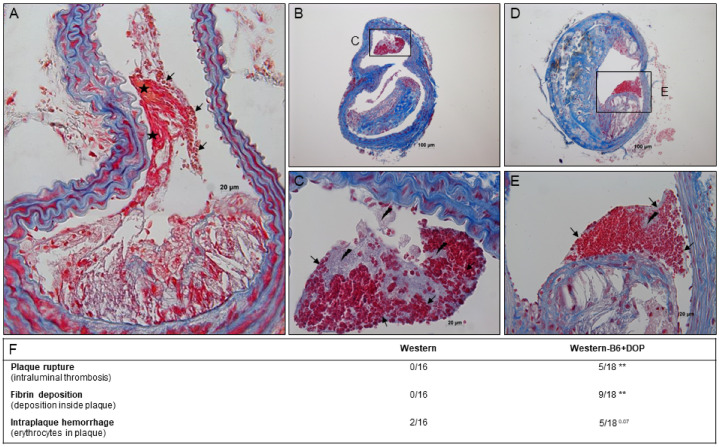
**Plaque rupture and intraluminal thrombosis in DOP-treated *ApoE^−/−^* mice: representative findings and quantitation.** (**A**–**E**) Representative BCA lesions from *ApoE^−/−^* mice on Western-B6+DOP for 12 weeks stained with Carstairs featuring different types of plaque rupture and thrombosis. (**A**) Fibrous cap break into the lumen with appositional thrombus consisting of erythrocytes (→) and fibrin (∗) that protrudes into the lumen (thrombus extends several hundred µm up and downstream of the rupture site). (**B**–**E**) Further examples of plaque rupture and intraluminal thrombosis associated with individual plaque structures, thrombus containing erythrocytes (→) and platelets (flash sign); (**C**) and (**E**) are insets from (**B**,**D**) at a higher magnification. Magnification 400× (**A**); 200× (**B**); 600× (**C**); 100× (**D**); 400× (**E**). (**F**) The groups were compared with a chi-square test. Plaque rupture (chi-square 9.874, *p* = 0.0072); fibrin deposition (chi-square 11.43, *p* = 0.0033); intraplaque hemorrhage (chi-square 5.349, *p* = 0.0689); ** *p* < 0.01.

**Figure 5 ijms-23-09606-f005:**
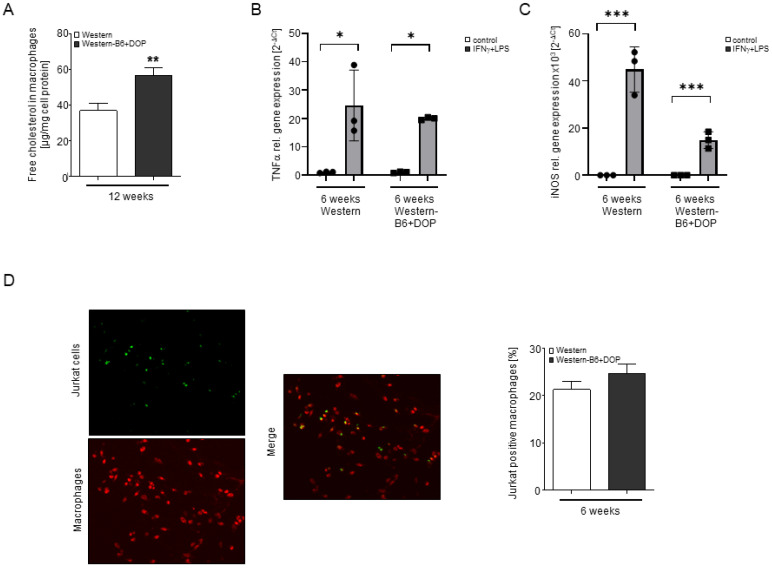
**High intracellular cholesterol but unaltered polarization, inflammation and efferocytosis after lyase inhibition.** (**A**) Free cholesterol content of resident peritoneal macrophages isolated from mice after 12 weeks of DOP treatment and their respective controls. Statistically significant differences from controls are indicated as ** *p* < 0.01 (Student’s *t*-test). (**B**,**C**) Quantification of TNF (**B**) and iNOS (**C**) gene expression by qRT-PCR analysis in thioglycollate-elicited peritoneal macrophages isolated from *ApoE^−/−^* mice on Western versus Western-B6+DOP diet for 6 weeks and stimulated or not in vitro with 1 μM LPS and 250 U IFN for 6 h. Data are presented as mean ± SEM. Statistical analysis was performed using a one-way ANOVA followed by a Tukey’s post hoc test. Statistically significant differences from controls are indicated as * *p* < 0.05, and *** *p* < 0.001. (**D**) Comparison of macrophage phagocytosis of apoptotic Jurkat cells between thioglycollate-elicited peritoneal macrophages from *ApoE^−/−^* mice fed a Western or Western-type-B6+DOP diet for 6 weeks. Representative images of macrophages, apoptotic Jurkat cells, overlay of both (left) and quantification (right). Results are shown as mean ± SEM. There were no statistically significant differences (Student’s *t*-test).

**Figure 6 ijms-23-09606-f006:**
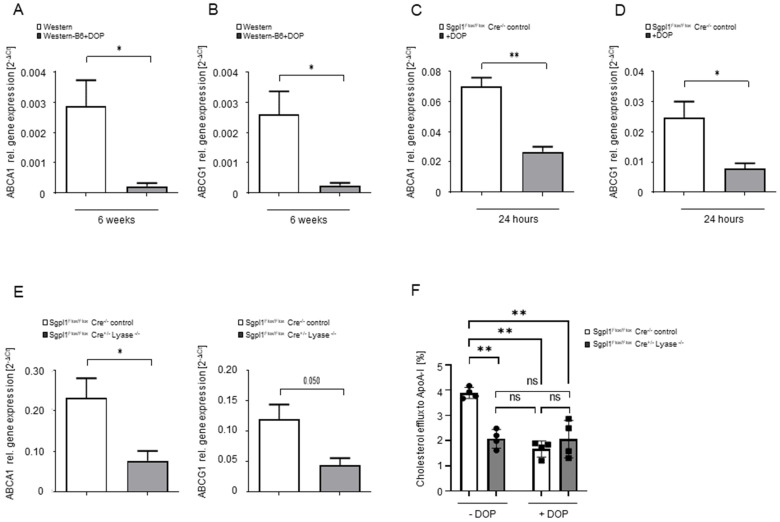
**S1P lyase inhibition and deficiency lead to ABCA1 and ABCG1 gene downregulation and reduced cholesterol efflux.** (**A**,**B**) Quantification of ABCA1 and ABCG1 gene expression by qRT-PCR analysis in thioglycollate-elicited peritoneal macrophages from mice treated or not with DOP for 6 weeks. (**C**,**D**) ABCA1 and ABCG1 gene expression in *Sgpl1^Flox/Flox^ Cre^−/−^* macrophages treated with 0.25 mM DOP or not for 24 h. (**E**) ABCA1 and ABCG1 gene expression in peritoneal macrophages from *Sgpl1^Flox/Flox^ Cre^−/−^* and *Sgpl1^Flox/Flox^ Cre^+/−^* mice. Statistically significant differences from controls are indicated as * *p* < 0.05 and ** *p* < 0.01 (Student’s *t*-test). (**F**) [^3^H]-cholesterol efflux to ApoA-I under basal conditions and after treatment with 0.25 mM DOP for 24 h in peritoneal macrophages from *Sgpl1^Flox/Flox^ Cre^−/−^* and *Sgpl1^Flox/Flox^ Cre^+/−^* mice. Statistical analysis was performed using a two-way ANOVA followed by a Tukey’s post hoc test. Statistically significant differences are indicated as * *p* < 0.05 and ** *p* < 0.01.

**Figure 7 ijms-23-09606-f007:**
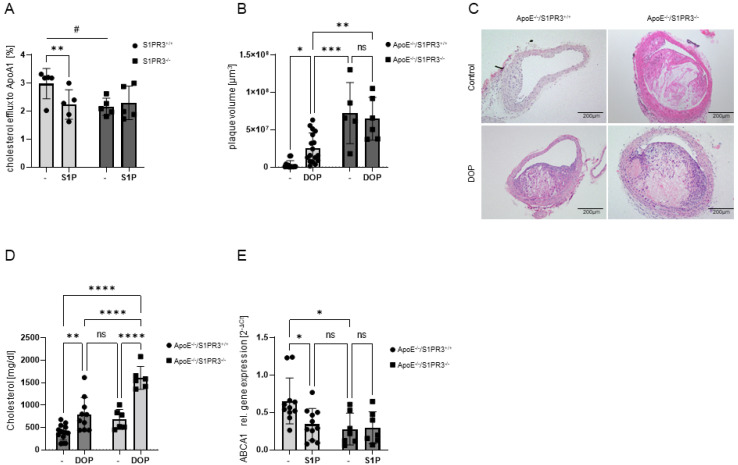
**Macrophage cholesterol efflux and ABCA1 gene expression, plasma cholesterol, and atherosclerosis in S1PR3 deficient mice.** (**A**) [^3^H]-cholesterol efflux to ApoA-I with and without 1 µM S1P added 30 min prior to ApoA-I and measured after 5 h in *S1PR^+/+^* and *S1PR3^−/−^* mice. (**B**) Quantification of plaque volume in the brachiocephalic artery after 12 weeks of Western versus Western-B6+DOP diet using Cavalieri’s stereologic method. (**C**) Representative images from (**B**). (**D**) Plasma cholesterol levels after 12 weeks of Western versus Western-B6+DOP diet in *ApoE^−/−^* and *ApoE^−/−^/S1PR3^−/−^* mice. (**E**) Macrophage ABCA1 gene expression with and without 1 µM S1P stimulation for 6 h in *ApoE^−/−^/S1PR^+/+^* and *ApoE^−/−^/S1PR3^−/−^* mice. Data in (**A**,**B**) are presented as mean ± SD and tested with stack matched two-way ANOVA (# for comparing genotypes; * for comparing treatments); # *p* < 0.05; ** *p* < 0.01; *** *p* < 0.001). Data in (**D**,**E**) are presented as mean ± SD and tested with two-way ANOVA; ns = not significant; ** *p* < 0.01; **** *p* < 0.0001.

**Table 1 ijms-23-09606-t001:** Altered sphingolipid content in macrophages and plasma following S1P lyase inhibition.

Macrophages [pmol/Cell × 10^6^]					Plasma [pmol/mL]	
	Western 1 wk	Western-B6+DOP 1 wk	Western 6 wk	Western-B6+DOP 6 wk	Western 6 wk	Western-B6+DOP 6 wk
**S1P**	0.48 ± 0.10	462.62 ± 18.36 **	0.17 ± 0.17	142.13 ± 31.46 ^§^	1411 ± 221	3854 ± 702 *
**Sph**	2.9 ± 0.9	268.3 ± 26.5 **	3.5 ± 0.6	51.5 ± 9.5 ^§^	138 ± 22	255 ± 48
**Cer-16**	42.7 ± 16.2	658.1 ± 94.7 *	77.8 ± 7.6	245.0 ± 29.7 ^§§^	3533 ± 1436	8031 ± 1504 *
**SM**	1139 ± 287	2589 ± 405 *	800 ± 125	737 ± 93	413,845 ± 168,542	414,410 ± 70,540
**PC**	282 ± 87	719 ± 96 *	510 ± 86	433 ± 60	225,378 ± 30,820	315,700 ± 14,504 *

Left, ApoE^−/−^ mice receiving Western-type-B6 for the indicated times were treated with or without DOP for 1 and 6 weeks (wk) and the concentrations of S1P, sphingosine (Sph), ceramide 16 (Cer-16), sphingomyelin (SM), and phosphatidylcholine (PC) were determined by liquid chromatography coupled to tandem mass spectrometry in thioglycollate-elicited peritoneal macrophages (n = 3, 3, 3, and 4). Right, concentrations of the same metabolites in plasma after 6 weeks (n = 5 and 6). Data are presented as means ± SEM. Statistically significant differences from the indicated controls were analyzed by Student’s *t-*test. *^§^ *p* < 0.05; **^,§§^ *p* < 0.01.

## Data Availability

Not applicable.

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
