# Peer review of "Sphingosine-1-Phosphate (S1P) Lyase Inhibition Aggravates Atherosclerosis and Induces Plaque Rupture in ApoE−/− Mice"

_ijms, 2022, doi:10.3390/ijms23179606_

Round 1

Reviewer 1 Report

The authors addressed the impact of persistent high levels of  S1P on atherosclerosis in ApoE-/- mice over 12 weeks,  by preventing the S1P degradation using  S1P lyase inhibitor  4-deoxy-14 pyridoxine (DOP). They showed that DOP treatment promoted atherosclerosis, compromised cholesterol efflux and caused genuine plaque rupture. Mechanistically, this group previously reported that S1PR3 is critically  involved in macrophage cholesterol efflux to ApoA-I, here they further tested if S1P/S1PR3 signaling regulated cholesterol efflux to ApoA-I. In vitro assay, the authors demonstrated that S1P lyase inhibition and deficiency lead to the downregulation of  peritoneal macrophage ABCA1 and ABCG1,  the cholesterol transporters. In vivo assay, they further showed a attenuated cholesterol efflux in S1PR3 deficient macrophages and enhanced atherosclerosis  in ApoE-/-/S1P3-/- mice. This study implicated a new function of macrophage S1P/S1PRs signaling in the pathogenesis of atherosclerosis,  by extending to macrophage cholesterol efflux.

Major concerns:

The authors failed to provide direct evidence of S1P/S1P3R signaling to ABCA1 and ABCG1 production in macrophages. This would be easily done by stimulating WT or S1PR1-3 KO macrophages with S1P at certain timepoint and measure ABCA1 and ABCG1 by RT-PCR.

Minor concerns:

1. In Figure 5B, 5C, and Figure 6A-F. Instead of fold changes, it would be better to show the relative gene expression  against  normalizing housekeeping genes like HPRT, GAPDH etc.

2. In Figure 7A, It would be better to indicate the mice as: “Wildtype” /”S1PR3-/+” and  “S1PR3-/-“. The same for Figure 7B: ApoE-/- and  ApoE-/-/S1PR3-/-.

Author Response

Thank you for the careful and insightful comments. Please find enclosed our response.

Reviewer 2 Report

The manuscript by Keul et al investigates the role of S1P in atherosclerosis and plaque rupture using ApoE-/- Mice. After administering DOP, a pharmacological inhibitor of the S1P lyase, to the mice, they showed that high levels of S1P accelerated the development of atherosclerosis and caused frequent plaque rupture with atherothrombosis. They furthermore demonstrated that high S1P levels decreased cholesterol efflux and downregulated the expression of ABCA1/G1 transporters. Additionally, the authors demonstrate that S1PR3 mediates S1P signaling. Overall, this study is well-designed and offers interesting information about the metabolism of sphingolipids and cholesterol.

Specific comments:

  1. In some figures, there are borders around the Y-axis. For example: Fig 5, Fig 6.
  2. Table 1. Could you explain why sphingolipid and cholesterol contents in macrophages from Western-DOP mice were lower at 6 weeks of age than they were at 1 week?
  3. Did you compare the cholesterol efflux of macrophages from the Western+DOP-treated mice to macrophages from the Western group?
  4. Page 4, line 129. ‘S1P lyase inhibition is known to induce characteristic changes in lymphocyte subpopulations due to its disruption of S1P gradients between plasma and tissues’. Did you detect a difference in cardiac lipids between Western+DOP and Western groups? 

Author Response

Thank you very much for the careful and helpful comments. We have addressed them below and have included all relevant data and aspects in the manuscript.

Round 2

Reviewer 2 Report

All the comments have been addressed in the newer version of the manuscript.

Minor:

Page 3, line 108. mice 'no' Western versus Western-B6+DOP diet

Page 11, line 277. geneexpression